# Strong biomechanical relationships bias the tempo and mode of morphological evolution

Martha M Muñoz[1,2]*, Y Hu[3], Philip S L Anderson[4], SN Patek[2]*

[1]Department of Biological Sciences, Virginia Tech, Blacksburg, United States; [2]Department of Biology, Duke University, Durham, United States; [3]Department of Biological Sciences, University of Rhode Island, Kingston, United States; [4]Department of Animal Biology, University of Illinois, Urbana-Champaign, United States

**Abstract** The influence of biomechanics on the tempo and mode of morphological evolution is unresolved, yet is fundamental to organismal diversification. Across multiple four-bar linkage systems in animals, we discovered that rapid morphological evolution (tempo) is associated with mechanical sensitivity (strong correlation between a mechanical system's output and one or more of its components). Mechanical sensitivity is explained by size: the smallest link(s) are disproportionately affected by length changes and most strongly influence mechanical output. Rate of evolutionary change (tempo) is greatest in the smallest links and trait shifts across phylogeny (mode) occur exclusively via the influential, small links. Our findings illuminate the paradigms of many-to-one mapping, mechanical sensitivity, and constraints: tempo and mode are dominated by strong correlations that exemplify mechanical sensitivity, even in linkage systems known for exhibiting many-to-one mapping. Amidst myriad influences, mechanical sensitivity imparts distinct, predictable footprints on morphological diversity.
DOI: https://doi.org/10.7554/eLife.37621.001

*For correspondence:
mmunoz5@vt.edu (MMM);
snp2@duke.edu (SNP)

**Competing interests:** The authors declare that no competing interests exist.

## Introduction

The uneven tempo of phenotypic evolution is a universal feature of biological systems, from proteins to whole-organism traits (*Simpson, 1944*; *Gingerich, 2009*; *Zhang and Yang, 2015*). Intrinsic and extrinsic mechanisms affecting rates of evolution have been probed extensively (*Wake et al., 1983*; *Gillooly et al., 2005*; *Eberhard, 2010*; *Zhang and Yang, 2015*). However, biomechanics – the intersection of mechanics and biology – is a key axis influencing phenotypic evolution (*Arnold, 1992*) that has been less often examined, and infrequently through the use of quantitative and comparative datasets (*Holzman et al., 2012*; *Wainwright et al., 2012*; *Collar et al., 2014*; *Muñoz et al., 2017*). Because rates of morphological divergence and speciation are often coupled (*Rabosky and Adams, 2012*), connecting biomechanics to morphological evolution enriches our understanding of the processes shaping diversification.

An enduring paradox in evolutionary biomechanics and functional morphology is whether strong correlations among traits (sometimes termed constraints) enhance or limit evolutionary diversification (*Gould, 1989*; *Antonovics and van Tienderen, 1991*; *Schwenk, 1994*). For example, strong morphological correlations could be predicted to reduce the rate or amount of morphological evolution, because even slight changes could compromise the system's proper functioning (*Raup and Gould, 1974*). Conversely, a strong correlation could enhance evolutionary change by providing a morphological pathway for adaptation (*Holzman et al., 2012*; *Muñoz et al., 2017*). A weak association could enhance the freedom to vary (*Collar et al., 2014*; *Schaefer and Lauder, 1996*), or,

**eLife digest** Imagine going for a swim on a shallow reef. You might see mantis shrimp striking at fish or snails, and reef fish gulping down smaller fish and plankton. Despite how different these movements are, rapid mantis shrimp strikes and fish suction are guided by the same mechanics: four-bar linkages. These shared mechanical systems evolved independently, much like the wings of birds and butterflies. Certain researchers study how organisms evolve based on biomechanics, the field of science that applies principles from mechanics to study biological systems.

Four-bar linkages, which are widespread in nature, consist of a loop made of four bars (or links) connected by four joints. The system allows a wide range of motions, and it is found anywhere from oil pumpjacks to the inside of the human knee. Researchers are interested in how similar mechanical systems like four-bar linkages influence the diversification of distantly related organisms, such as fish and crustaceans.

Changes in an element of a four-bar linkage can have widely different consequences because of a phenomenon known as mechanical sensitivity. Modifications of highly mechanically sensitive parts will have a dramatic effect on the system, while alterations in other areas have little or no effect.

Whether the most mechanically sensitive parts evolve faster or slower than the less sensitive elements is still up for debate. Changes in the sensitive elements could be severely constrained because these modifications may compromise the survival of the organisms. However, they could also help species adapt quickly to new environments. So far, researchers have found that in the four-bars linkage of the mantis shrimp, the most mechanically sensitive parts evolve the fastest. Yet, it was unclear whether this would also apply to other species.

Here, Muñoz et al. compared four-bar linkages in three families of fish and in mantis shrimp, and discovered that the most mechanically sensitive elements are the smallest links. These can undergo changes in length that have a strong impact on how the linkage works. In addition, evolutionary analyses showed that the most mechanically sensitive parts do indeed evolve the fastest in both mantis shrimp and fish. More work is now required to see if this pattern holds across various organisms, and if it can be considered as a general principle that drives evolution.

DOI: https://doi.org/10.7554/eLife.37621.002

conversely, weaken, and thereby reduce, the pathways to morphological change (*Alfaro et al., 2005*; *Collar and Wainwright, 2006*). Therefore, depending on the context, strong and weak correlations have been construed to enhance or restrict evolutionary diversification. A comparative approach can empirically resolve this conundrum, for example, by comparing rates and phylogenetic patterns of morphological evolution in similar, independently-evolved systems.

Here, we leverage the multiple independent evolutionary origins of four-bar linkage systems (*Figure 1*) to test how morphological and mechanical correlations impact two key aspects of morphological evolution: tempo (rate at which morphological disparity accumulates) and mode (evolutionary pattern of trait shifts across phylogeny). Four-bar linkages are closed-chain systems comprised of four rigid links that rotate to transmit motion and force (*Anker, 1974*; *Muller, 1996*; *Westneat, 1990*; *Martins, 1994*) (*Figure 1A*). Mechanical output of four-bar linkages is often measured in terms of kinematic transmission (KT), a simple metric that can quantify a tradeoff between displacement and force across different linkage configurations (see Materials and methods section for further information about KT). Four-bar linkages are widespread in nature, and enable a rich diversity of behaviors in vertebrates and invertebrates (*Westneat, 1990*; *Wainwright et al., 2005*; *Patek et al., 2007*; *Olsen and Westneat, 2016*). For example, four-bar linkages actuate mouth opening for suction feeding in fishes (*Westneat, 1990*; *Martins, 1994*; *Muller, 1987*), flexion and extension of the vertebrate knee (*Hobson and Torfason, 1974*), rapid strikes of the stomatopod raptorial appendage (*Patek et al., 2004*; *Patek et al., 2007*; *McHenry et al., 2012*; *McHenry et al., 2016*), and skull kinesis in birds (*Hoese and Westneat, 1996*).

Four-bar linkages, like any system built of three or more parts, exhibit many-to-one mapping, meaning that similar mechanical outputs (e.g. KT) can be produced through different combinations of morphology (e.g. link lengths; *Wainwright et al., 2005*; *Wainwright, 2007*). Four-bar linkages also exhibit mechanical sensitivity, which occurs when KT is disproportionately sensitive to variation

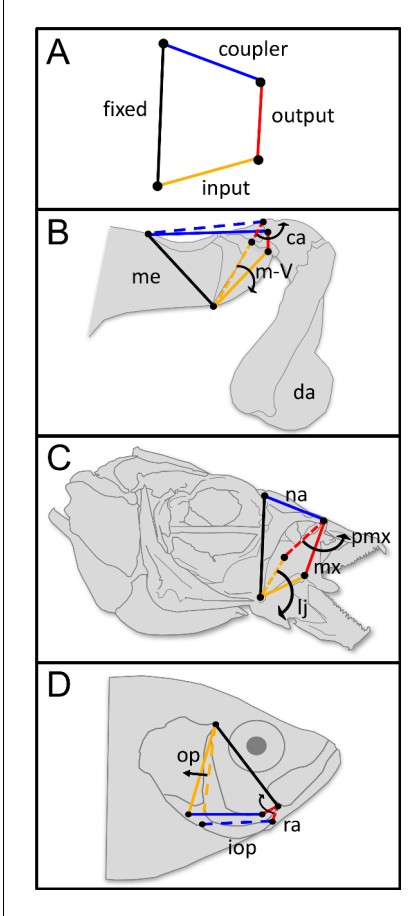

**Figure 1.** Four-bar linkage systems have evolved independently multiple times across animals and are comprised of four rotatable links that transmit motion and force. (**A**) Four-bar linkages consist of a fixed link (black) and three mobile links: input (orange), output (red), and coupler (blue). (**B**) In the raptorial appendages of mantis shrimp (Stomatopoda), rotation of the input link (meral-V, m-V, which is part of the merus segment, me) causes the output link (carpus, ca) to rotate outward, which then rapidly rotates the dactyl (da). (**C**) In the oral four-bar system that evolved independently in labrid and cichlid fishes, the input link (lower jaw, lj) rotates ventrally, causing rotation in the nasal (na) and in the output link (maxilla, mx), resulting in premaxillary (pmx) protrusion. (**D**) In the opercular four-bar linkage system of centrarchid fish, the input link (opercle and subopercle, op) swings posteriorly, as does the interopercle (iop). This motion is transmitted to the output link (retroarticular process in the mandible, ra), which causes the mandible to rotate ventrally and open the lower jaw. (**B–D**) Dashed lines denote the closed configuration of input (orange), output (red), and coupler (blue) links, whereas solid lines denote their open configuration following motion. Arrows denote the direction of motion. Distal/anterior is to the right and dorsal is toward the top of the page.
DOI: https://doi.org/10.7554/eLife.37621.003

in some links and relatively insensitive to variation in others (*Anderson and Patek, 2015*; *Muñoz et al., 2017*). Therefore, four-bar linkages exhibit (1) both weak and strong correlations among parts and outputs and (2) multiple evolutionary origins across the Metazoa. As such, four-bar linkages are a fertile testing ground for comparative analyses of evolutionary biomechanics and morphology.

To our knowledge, the connection between mechanical sensitivity and morphological evolution has only been studied in the four-bar linkage system of the mantis shrimp (Stomatopoda) raptorial appendage (*Muñoz et al., 2017*; *Anderson and Patek, 2015*) (*Figure 1*). The mantis shrimp four-bar system is used for feeding, fighting, and substrate manipulation via extremely rapid strikes of their raptorial appendages (*Patek et al., 2004*; *Patek et al., 2007*; *Patek et al., 2013*; *Patek and Caldwell, 2005*; *McHenry et al., 2016*; *deVries et al., 2012*; *Green and Patek, 2015*; *Green and Patek, 2018*; *Crane et al., 2018*). Spearing stomatopods that harpoon mobile, soft-bodied prey have higher KT (greater displacement) than smashing mantis shrimp that bludgeon hard-shelled prey using linkages with lower KT (greater force) (*Anderson et al., 2014*; *McHenry et al., 2016*). In mantis shrimp, mechanical sensitivity is associated with accelerated evolution: the link most tightly correlated with KT exhibits the fastest rate of evolution (*Muñoz et al., 2017*).

Tantalizing patterns often occur in individual clades or mechanical systems, yet few are robust to tests in multiple lineages. Therefore, previous findings in mantis shrimp leave uncertain whether the enhanced evolutionary rate associated with mechanical sensitivity is a system-specific finding or instead occurs across an array of taxa and reflects a more general pattern. Furthermore, the initial studies in mantis shrimp were analyses of evolutionary tempo, but the phylogenetic pattern of these rate changes (mode) was not analyzed. Analysis of mode can resolve whether phylogenetic shifts to a higher or lower KT are exclusively accompanied by shifts in link(s) to which KT is most sensitive, or occur through different morphological pathways that are not necessarily tied to mechanical sensitivity.

Here we examine rates of morphological evolution (tempo) in four particularly well-studied systems (*Figure 1*): (1) oral four-bar linkages of 101 species of wrasses (Family: Labridae) and (2) oral four-bar linkages of 30 species of cichlids (Family: Cichlidae), (3) opercular four-bar systems in 19 species of sunfish (Family: Centrarchidae)

and (4) the previously-published dataset in mantis shrimp that was analyzed using the same methods as in this study. In fish, the oral four-bar system actuates the upper jaw and the opercular four-bar actuates the lower jaw (*Figure 1*); together they open a large space in the mouth that creates negative pressure to suction prey (*Westneat, 1990*; *Martins, 1994*). Similarly to mantis shrimp (*Anderson et al., 2014*; *Anderson and Patek, 2015*), the mechanical tradeoffs between displacement and force represented by KT appear to generally track fish trophic ecology (*Wainwright et al., 2004*; *Hulsey and Garcia De Leon, 2005*). For example, fish that pursue elusive prey tend to have oral four-bar linkages with higher KT (resulting in greater displacement for snagging rapid prey), whereas those that scrape algae tend to have lower KT (resulting in greater force, such as for dislodging sessile, encrusted food items from hard surfaces) (*Hulsey and Garcia De Leon, 2005*). Finally, we examine the phylogenetic pattern (mode) of shifts in KT and links across the especially well-sampled oral four-bar system in wrasses, to test how KT and link lengths change across the phylogeny and whether these changes occur concordantly as predicted by mechanical sensitivity.

## Results

We estimated the Brownian motion evolutionary rate parameter, $\sigma^2$ (bounded by its 95% confidence interval), which represented the net rate of phenotypic change over time (*Felsenstein, 1985*; *Martins, 1994*; *O'Meara et al., 2006*) for the three mobile links – input, output, and coupler – of each four-bar system (See Materials and ethods). A single consistent result emerged from our analysis of evolutionary rates: stronger correlations between link lengths and kinematic transmission (KT) were associated with faster rates of morphological evolution. In each system, we found that mechanical sensitivity was always associated with a faster rate of link evolution (*Figure 2*; *Supplementary file 1*). Evolutionary rate can be artificially inflated by greater trait variance (*O'Meara et al., 2006*; *Adams, 2013*); we incorporated intraspecific measurement error into our rate estimates and confirmed that a higher evolutionary rate was not driven by greater variance (*Supplementary file 2*).

Even though rates of morphological evolution consistently tracked mechanical sensitivity, the particular links associated with mechanical sensitivity differed across the four-bar systems (*Supplementary files 3–6* [rotatable 3D phylomorphospace plots]; *Table 1*). For example, in the cichlid oral four-bar system, mechanical output was positively correlated with input link length (PGLS $r^2$=0.62, $p<10^{-6}$), inversely correlated with the coupler link length ($r^2$=0.27, $p$=0.002), and exhibited no relationship with the output link length (*Supplementary file 3*; *Table 1*). By contrast, in three systems - the wrasse oral four-bar, the sunfish opercular four-bar, and the stomatopod raptorial four-bar - the output link length was a strong predictor of mechanical output (PGLS $r^2$>0.66, $p<10^{-11}$), whereas the coupler link only weakly predicted mechanical output (PGLS $r^2$<0.14) (*Supplementary files 4–6*). The oral four-bars of cichlids and wrasses share a common evolutionary origin (*Alfaro et al., 2004*); nonetheless, rate differences were predicted by mechanical sensitivity rather than shared ancestry. Hence, analogous four-bar systems do not result in common patterns of mechanical sensitivity, whereas mechanical sensitivity is consistently a strong predictor of evolutionary rate differences.

We next examined the phylogenetic pattern of trait shifts in KT and each mobile link in the wrasse oral four-bar system. We applied a Bayesian framework in the program bayou (*Uyeda and Harmon, 2014*) to reconstruct phylogenetic shifts in the Ornstein-Uhlenbeck (OU) optimal trait parameter ($\theta$) for morphological components and mechanical output. The OU model of evolution is characterized by the presence of an adaptive peak, with the peak representing the optimal value for a given trait. Thus, $\theta$ reflects the evolutionary optimal trait value as inferred from an OU-process on the phylogeny. To be clear for interdisciplinary readers, $\theta$ is not a metric for calculating a biomechanically optimal trait for a certain mechanical function.

By estimating $\theta$ across the wrasse phylogeny, we pinpointed the nodes associated with strongly supported shifts to higher or lower values in KT and link size. We performed this analysis only on the largest and most species-rich dataset (wrasses: >100 species sampled), because evolutionary inferences are unstable with fewer than 50 taxa (*Uyeda and Harmon, 2014*). We detected three well-supported evolutionary shifts in KT (posterior probability [pp] range 0.65–0.99; *Supplementary file 7*) (*Figure 3*). For each of these shifts in mechanical output, we also detected strongly supported shifts in the output and input links, but never the coupler link (*Figure 3*; *Figure 3—figure supplement 1*). Therefore, the three shifts in KT occur through three different morphological pathways, but

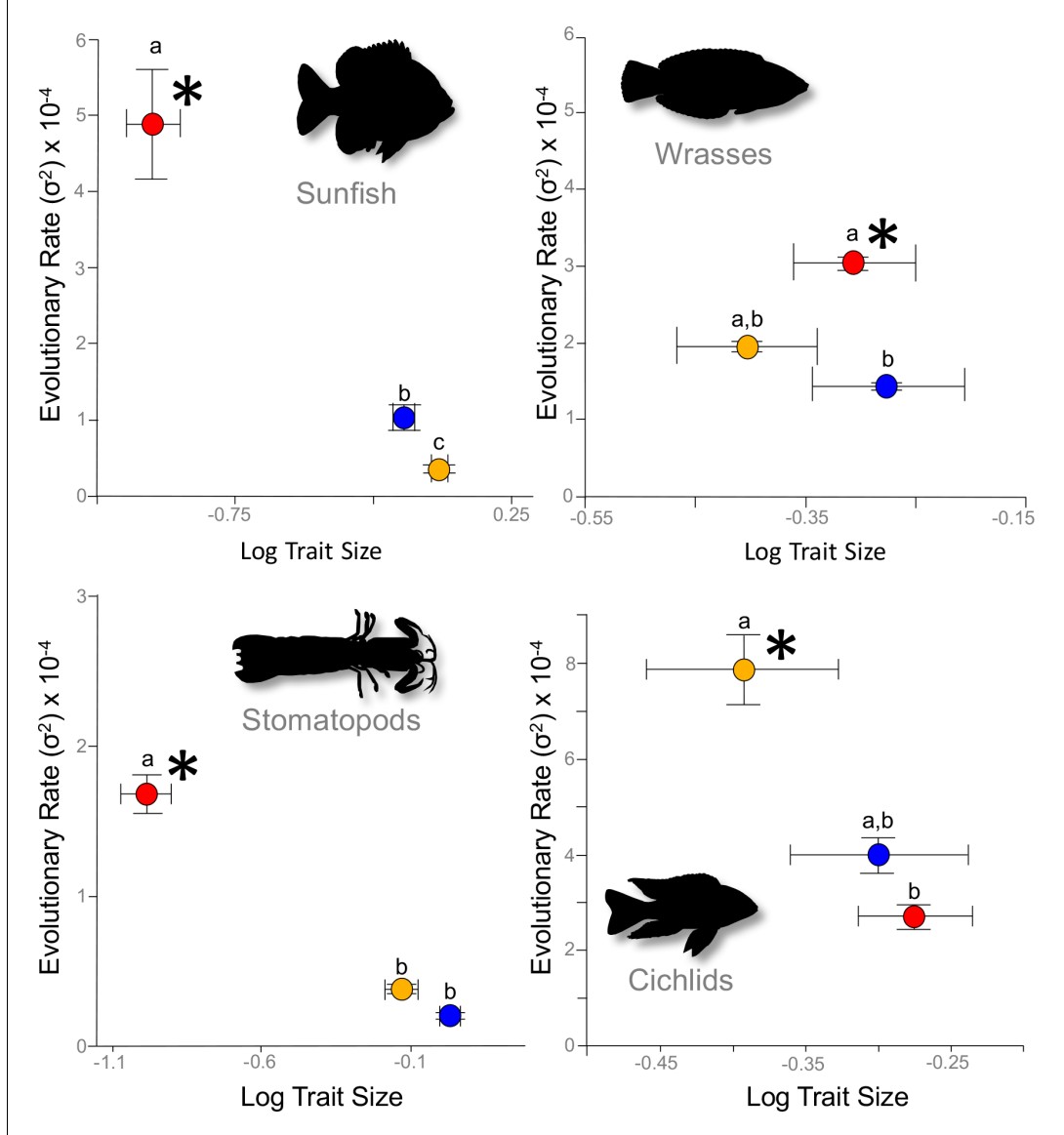

**Figure 2.** Across the four focal systems, evolutionary rate is consistently faster in the links to which the mechanical output is most mechanically sensitive (asterisks). The evolutionary rate parameter, $\sigma^2$ (± 95% confidence interval), is depicted for each link in each system. Orange circles denote the input link, red circles denote the output link, and blue circles denote the coupler link. Shared letters denote rates that are not statistically different from each other (statistical results are in *Table 1*).

DOI: https://doi.org/10.7554/eLife.37621.004

only via the changes in the links to which KT is most mechanically sensitive. The evolutionary shifts to higher mechanical output (increased KT) occur twice – once in razorfishes (pp = 0.67) and once in the branch leading to the Creole wrasse, *Clepticus parrae* (pp = 0.99). In razorfishes, the transition to higher KT is accompanied by a shift to a smaller output link (pp = 0.88), and in Creole wrasse, by both an increase in input link length (pp = 0.97) and a reduction in output link length (pp = 0.77). A transition to lower KT in the *Anampses* clade is accompanied by a reduction in input link length (pp = 0.95), with no concomitant shifts in output link length.

**Table 1.** Phylogenetic Generalized Least Squares (PGLS) regressions reveal that the relationship between KT and link size (mechanical sensitivity) varies among four-bar linkage systems.
In each analysis, the mobile links are predictor variables and kinematic transmission is the response variable.

**Cichlids (df = 28)**

| Predictor | Coeff.±SE | T | P | r² | AIC |
|---|---|---|---|---|---|
| Input | 0.63 ± 0.09 | 7.08 | 1.1e-7 | 0.623 | −119.22 |
| Output | −0.23 ± 0.24 | −0.96 | 0.343 | 0.002 | −89.77 |
| Coupler | −0.46 ± 0.14 | −3.42 | 0.002 | 0.270 | −98.95 |

**Mantis shrimp (df = 34)**

| Predictor | Coeff.±SE | T | P | r² | AIC |
|---|---|---|---|---|---|
| Input | 0.20 ± 0.32 | 0.64 | 0.527 | 0.017 | −90.86 |
| Output | −0.77 ± 0.08 | −9.75 | 2.3e-11 | 0.729 | −138.34 |
| Coupler | 0.03 ± 0.39 | 0.07 | 0.948 | 0.029 | −90.48 |

**Sunfish (df = 17)**

| Predictor | Coeff.±SE | T | P | r² | AIC |
|---|---|---|---|---|---|
| Input | 3.02 ± 0.68 | 4.48 | 3.3e-4 | 0.514 | −44.72 |
| Output | −1.18 ± 0.04 | −26.56 | 2.7e-15 | 0.975 | −97.92 |
| Coupler | −1.00 ± 0.63 | −1.58 | 0.133 | 0.133 | −32.48 |

**Wrasses (df = 99)**

| Predictor | Coeff.±SE | T | P | r² | AIC |
|---|---|---|---|---|---|
| Input | 1.00 ± 0.13 | 7.45 | 3.5e-11 | 0.353 | −240.36 |
| Output | −1.47 ± 0.10 | −14.15 | 2.2e-16 | 0.666 | −306.55 |
| Coupler | 0.39 ± 0.18 | 2.25 | 0.027 | 0.039 | −201.14 |

DOI: https://doi.org/10.7554/eLife.37621.005

## Discussion

Amidst the morphological, behavioral and ecological diversity of four-bar linkages, in every system we tested, greater mechanical sensitivity is associated with faster morphological evolution. The connection between mechanical sensitivity and evolutionary rate is therefore robust to independent origins and distinct behavioral functions, suggesting a generalizable phenomenon in four-bar linkage systems. These findings address a longstanding conundrum of constraints in evolution (discussed in *Gould, 1989*; *Antonovics and van Tienderen, 1991*; *Schwenk, 1994*) – specifically, whether strong correlations among traits should enhance or reduce evolutionary change – by demonstrating that strong correlations between components and mechanical output accelerate evolutionary change (*Figure 4*). Our results further reveal how many-to-one mapping and mechanical sensitivity enable multiple configurations while simultaneously biasing those configurations to a subset of traits.

Both absolute and relative link sizes influence evolutionary rates, and the structural geometry of four-bar linkages is central to understanding these findings (*Muller, 1996*). In terms of absolute size, the same length change applied to small and large links is proportionally larger for the small link, such that KT is most influenced by the change to the small link. Therefore, changes to the smallest links induce disproportionately large changes in the system's geometry and, therefore, the transmission of force and motion (*Anderson and Patek, 2015*; *Hu et al., 2017*). In terms of relative size, greater disparity among link sizes in a four-bar system results in greater evolutionary rate disparity. These effects of size are especially apparent in the stomatopod raptorial four-bar linkage and sunfish opercular four-bar linkage. In both of these systems, the output link is approximately an order of magnitude smaller than the input and coupler links, and it also exhibits an order of magnitude faster evolution (*Supplementary file 1*). In contrast, relative link lengths vary less dramatically in the oral four-bar system of cichlids and wrasses, and the corresponding evolutionary rate shifts are

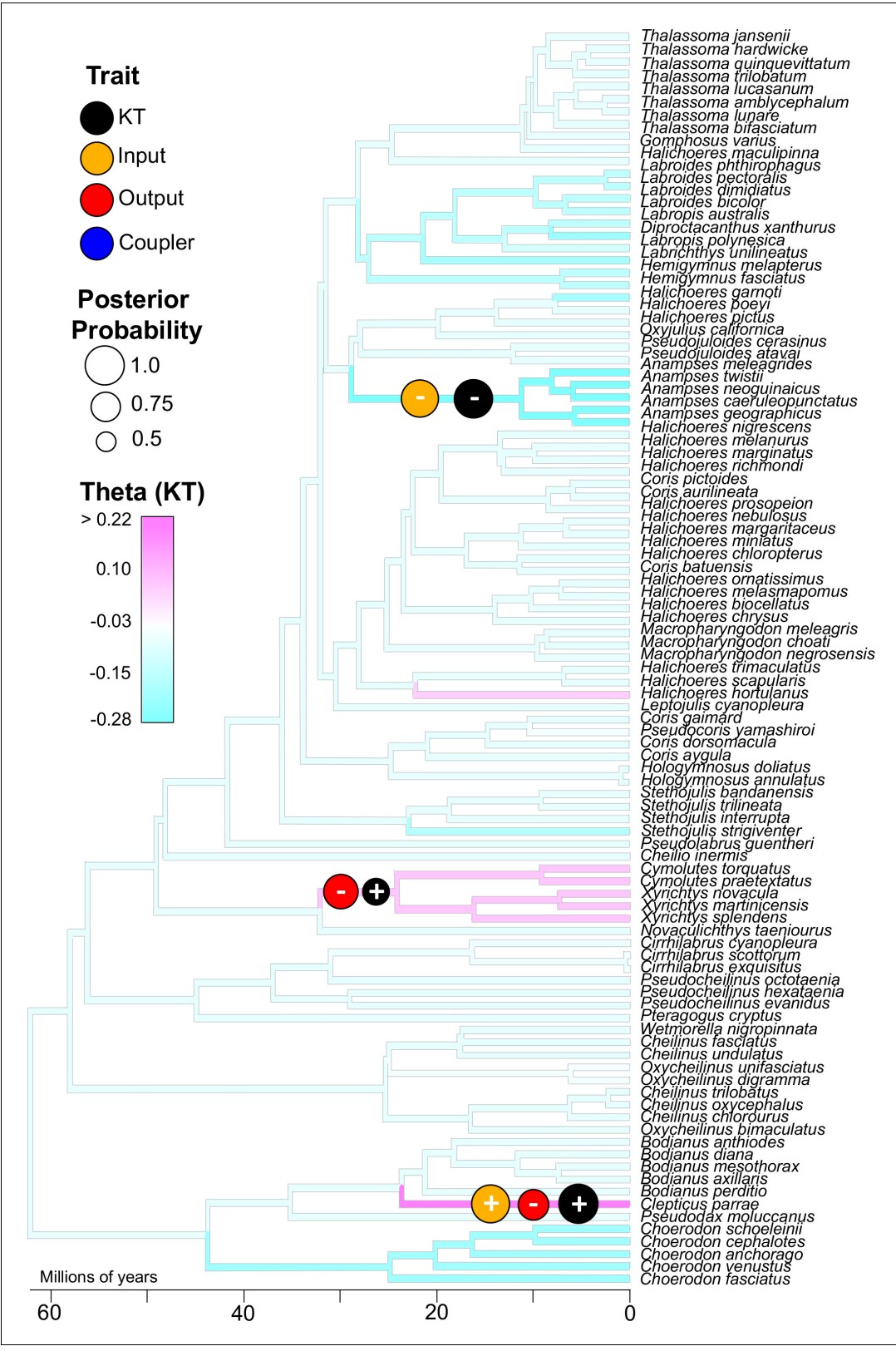

**Figure 3.** Three well-supported transitions in KT (black circles) occurred across the phylogeny of wrasses. With each of these transitions in KT, either the input (orange circle) or output link (red circle) also experienced a strongly-supported shift in magnitude (sign indicates directionality of trait shift). The coupler link (blue circle) did not exhibit a strongly-supported shift. These analyses were performed using reversible-jump MCMC which detected significant shifts (optimal trait value, θ) based on the distribution of traits across the phylogeny (KT trait distribution is overlaid as a color map on the

*Figure 3 continued on next page*

*Figure 3 continued*

tree branches; see *Figure 3—figure supplement 1* for color maps of the other trait distributions). The sizes of the circles represent posterior probability (threshold posterior probability for a strongly supported transition was set at > 0.5.

DOI: https://doi.org/10.7554/eLife.37621.006

The following figure supplement is available for figure 3:

**Figure supplement 1.** Results of reversible-jump MCMC analysis of evolutionary trait shifts for the input link (**A**), output link (**B**), coupler link (**C**), and KT (**D**).

DOI: https://doi.org/10.7554/eLife.37621.007

statistically weaker, although still exhibiting two- to four-fold rate differences (*Figure 2*; *Supplementary file 1*).

Biological sizes, whether of genomes, cells, or the organisms themselves, help sculpt macroevolutionary dynamics by influencing patterns of trait evolution or shaping deeper-scale patterns of lineage diversification (*Hanken and Wake, 1993*; *Uyeda et al., 2017*). Here, link size plays a central role in determining the physical basis for mechanical sensitivity and evolutionary rate disparity, indicating that size-scaling relationships in biomechanics can mediate evolutionary dynamics. Although we connect mechanical sensitivity to a relatively faster rate of evolution, an interesting next step is to disentangle whether such traits evolve more quickly because of strong directional selection on small links or whether the other, relatively larger traits of the system evolve more slowly due to stabilizing selection (*Arnold, 1983*; *Arnold, 1992*). Statistically comparing these two possibilities requires especially broad sampling: in the growing age of big data in digital morphology and phylogenetics, this task is rapidly becoming feasible (*Davies et al., 2017*).

Our analysis of the phylogenetic pattern (mode) of trait evolution exemplifies the integration of mechanical sensitivity and many-to-one mapping. An implicit assumption of many-to-one mapping is freedom of evolution: theoretically, any alternative configurations yielding a similar mechanical output should be equally likely to evolve (*Wainwright et al., 2005*; *Wainwright, 2007*). Our findings in wrasses, however, demonstrate that mechanical sensitivity biases evolutionary transitions to traits with the greatest influence on mechanical output (input and output links). For each of the three

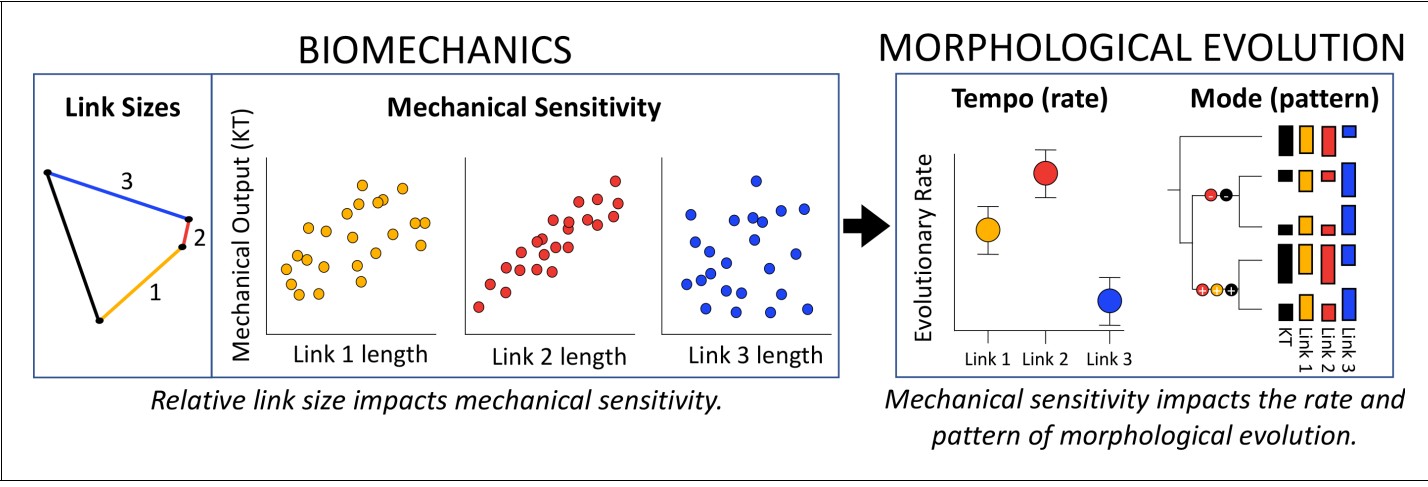

**Figure 4.** The three conceptual frameworks of many-to-one mapping, mechanical sensitivity and constraints converge at the intersection of biomechanics and morphological evolution, highlighting the importance of analyzing these systems at multiple levels. Many-to-one mapping (multiple configurations yield similar mechanical outputs) can occur in any linkage system, yet mechanical sensitivity defines the tight correlations between link size and kinematic transmission (KT). From a biomechanical perspective, mechanical sensitivity occurs because length changes in short links have disproportionately large effects on KT. From an evolutionary perspective, rate of evolutionary change (tempo) is accelerated in the links to which KT is most mechanically sensitive. Many-to-one mapping again emerges from analyses of evolutionary mode (pattern), in which statistically significant shifts in link length and KT across the topology of the phylogeny can occur through multiple configurations, yet only in the links to which the system is most mechanically sensitive.

DOI: https://doi.org/10.7554/eLife.37621.008

major KT shifts in wrasses, we detected three distinct morphological pathways involving either the input link, output link, or both (but never the coupler link). Correspondingly, both the output link and input link are strong predictors of KT in the wrasse oral four-bar, and the coupler is a weak predictor of KT. Among these links, there is some redundancy, as evidenced by the various morphological pathways accompanying transitions in KT. Therefore, mechanical sensitivity restricts the freedom of evolution central to many-to-one mapping by biasing evolutionary transitions to traits with the strongest effect on mechanical output.

When considered in the broader context of ecology and behavioral function, our findings raise as many questions as they answer. For example, if the shortest link is the output link, an animal achieves greater output rotation of the system than they input. Mantis shrimp use their shortest link (output link) to dramatically amplify an approximately nine-degree rotation of the input link; this enables the notoriously fast angular velocity of their predatory appendages (*Patek et al., 2004*,*Patek et al., 2007*; *Cox et al., 2014*; *McHenry et al., 2016*). In the centrarchids, the input rotation is controlled by a muscle too small to generate a large gape: instead they use linkage geometry to amplify gape (*Durie and Turingan, 2004*; *Camp and Brainerd, 2015*; *Camp et al., 2015*). In both cases, the location of the smallest link as the output link is directly related to the behavioral and ecological function of the linkage mechanism. Do small links participate in fewer or more mechanical functions than the larger links? Does selection favor the evolution of small linkages as pathways for strong mechanical sensitivity? Are macroevolutionary shifts in linkage geometry concordant with changes in diversification rate? In the case of the oral four-bar of teleosts, the maxilla (output link) is correlated with jaw protrusion, a key ecological aspect of fish diversification (*Hulsey and Garcia De Leon, 2005*). As such, the most mechanically influential links may play a crucial role in shaping diversification during speciation and adaptive radiation.

Perhaps the most imperative message of these findings is the necessity for multiple levels of analysis (*Jablonski, 2017a*;*Jablonski, 2017b*). Simply demonstrating many-to-one mapping in biomechanical systems is likely to miss a rich suite of evolutionary and mechanical dynamics that shape diversification (*Figure 4*). By considering the tempo and mode of evolution, generalizable interactions between mechanics and diversification are likely to emerge (*Scales and Butler, 2016*; *Blanke et al., 2017*). Evolutionary studies of many-to-one mapping have flourished in recent years, with particular focus on how natural selection and developmental constraints impact the evolution of complex mechanical systems in the wild (e.g. *Martinez and Sparks, 2017*; *Moody et al., 2017*; *Thompson et al., 2017*). Explicit consideration of mechanical sensitivity can enhance these studies by providing a general blueprint of the mechanical and evolutionary expectations for phenotypic diversification.

Deciphering the abiotic and biotic factors impacting the tempo and mode of phenotypic diversification is a topic of perennial interest and debate (*Simpson, 1944*; *Mayr, 1963*; *Gould, 1977*; *Gould, 2002*; *Schluter, 2000*; *West-Eberhard, 2003*; *Arnold, 2014* and references therein). Our findings reveal that mechanical sensitivity impacts macroevolutionary dynamics and point toward general rules connecting biomechanics and morphological diversification (*Figure 4*). Mechanical sensitivity is consistently associated with increased rates of evolutionary change, and the biomechanical impacts of link length provide a proximate explanation for the association between four-bar mechanics and the tempo of evolutionary change. Whereas many-to-one mapping in four-bar linkages theoretically supplies multiple morphological pathways for mechanical variation (*Alfaro et al., 2004*; *Alfaro et al., 2005*; *Wainwright, 2007*), mechanical sensitivity influences the tempo and mode of how these pathways evolve. This conclusion is relevant beyond mechanical systems. Many-to-one mapping is widespread in biomechanics (*Wainwright et al., 2005*) and can be applied even more broadly to assess various hierarchical phenomena in biology, such as molecular and cellular processes, including epistatic genotype-phenotype interactions (*Phillips, 2008*), physiological adaptations (*Scott, 2011*; *Cheviron et al., 2014*), and organism-level phenomena, including neurophysiological processes governing behavior (*Katahira et al., 2013*; *York and Fernald, 2017*). Extending this framework within and beyond biomechanics may yield a predictive understanding of the tempo and mode of evolutionary diversification.

## Materials and methods

### Four-bar linkage mechanics

In the teleost oral four-bar system, the lower jaw (coronoid process - mandible-quadrate joint) functions as the input link and its rotation relative to the fixed link (suspensorium - neurocranium) causes outward rotation in the maxilla (output link), resulting in premaxillary protrusion (*Westneat, 1990*; *Martins, 1994*). The teleost opercular four-bar linkage system facilitates feeding through lower jaw movement. In this case, the operculum (input link) rotates relative to the suspensorium (fixed link). This rotation transfers through the interopercle bone and the interopercle-mandible ligament (coupler link), and then to the retroarticular process of the mandible (output link). Rotation of the output link actuates jaw depression, which also contributes to the motion of the oral four-bar system that facilitates feeding via protrusion of the premaxilla (*Westneat, 1990*; *Martins, 1994*). During a mantis shrimp strike, the input link (meral-V) rotates distally, relative to the rest of the merus (fixed link) (*Patek et al., 2007*). This distal rotation releases the stored elastic energy, and pushes the carpus segment (output link), causing it to rotate and actuate the swinging arm. The coupler of the system is an extensor muscle that remains contracted during the strike.

### Morphological, mechanical, and phylogenetic data

Multispecies data from four-bar linkage systems were gathered from previous studies of the raptorial four-bar of mantis shrimp (Order: Stomatopoda), the oral four-bar of wrasses (Family: Labridae) and cichlids (Family: Cichlidae), and the opercular four-bar of sunfish (Family: Centrarchidae). The wrasse morphological and biomechanical dataset (*Alfaro et al., 2004*) is comprised of 101 species from 32 genera that we pruned down from 122 species using a recent time-calibrated phylogeny (*Baliga and Law, 2016*). Topology, branch lengths, and divergence times were estimated in a Bayesian framework using a relaxed clock model approach, with both mitochondrial and nuclear genes, and six fossils providing calibration points. The cichlid morphological and biomechanical dataset (*Hulsey and Garcia De Leon, 2005*) is comprised of 30 species from 13 genera, which we pruned down from the 97 species represented in a recent time-calibrated mitochondrial phylogeny (*Hulsey et al., 2010*). The sunfish dataset (*Hu et al., 2017*) consists of morphological and biomechanical data for 19 species from eight genera. The time-calibrated centrarchid phylogeny was estimated using both mitochondrial and nuclear DNA and six fossil calibration points (*Near et al., 2005*). The stomatopod morphological and biomechanical dataset (*Anderson and Patek, 2015*) consisted of 36 species from six superfamilies. We used a time-calibrated phylogeny (*Porter et al., 2010*), which we pruned from 49 species to the 36 analyzed in this study. The tree was constructed using both mitochondrial and nuclear genes, and fossil data defined the calibration points. Phylogenies and tables with raw data are provided in *Supplementary files 8–15*.

### Strengths and limitations of kinematic transmission

Kinematic transmission (KT) is the most widely used and readily available metric for characterizing the mechanical output of four-bar linkages (*Olsen and Westneat, 2016*). KT is calculated as the ratio of angular output motion relative to angular input motion (*Hulsey and Wainwright, 2002*). All else being equal, KT reflects a tradeoff between displacement and force: a higher KT yields greater displacement through more output rotation relative to input rotation, whereas lower KT yields greater force that occurs at the expense of displacement (*Westneat, 1994*). A strength of dimensionless metrics like KT is that they allow comparison across diverse groups of organisms. Furthermore, KT is calculated from the angles of rotation during motion and not from the linkages themselves (unlike, for example, mechanical advantage), such that examining the relationship between mechanical output and morphology is not autocorrelative.

Though widely applied and useful in various contexts, there are limitations to the robustness of KT as a mechanical metric. Recent work demonstrates that non-planar motion occurs in some biological linkage systems and that those systems should be studied as three-dimensional mechanisms (*Olsen and Westneat, 2016*; *Olsen et al., 2017*). The effect of non-planar movement on calculations of KT varies among systems and is unlikely to be a major source of error in the systems examined here (*Patek et al., 2007*; *McHenry et al., 2012,McHenry et al., 2016*; *Anderson et al., 2014*). Another issue is that KT is dynamic, meaning that its magnitude changes during the rotation of the

input link (*Patek et al., 2007*). In two of the fish systems examined (oral four-bar in wrasses and cichlids), KT was measured statically, specifically with the input link fixed at a starting angle of 30° in the lower-jaw rotation. This angle was chosen because it is biologically relevant for fish feeding (*Westneat, 1990*), and because starting angle was less important than link length for determining KT (*Wainwright et al., 2005*). In the sunfish and the mantis shrimp, KT was calculated in a dynamic fashion, by measuring the minimum value of KT over the course of its full rotation (*Anderson and Patek, 2015*; *Hu et al., 2017*).

To assess the effect of static versus dynamic methods for measuring KT, we re-analyzed the previously-collected four-bar linkage data for sunfish and mantis shrimp and calculated static KT measurements. Minimum KT measured previously on sunfish and mantis shrimp was determined by calculating instantaneous KT at every 0.1° of input rotation and using the minimum value found over the entire course of four-bar rotation (*Anderson and Patek, 2015*; *Hu et al., 2017*). To convert this dynamic measure to a static KT comparable to those in the wrasse and cichlid datasets, we averaged instantaneous KT over a specified overall input link rotation. For mantis shrimp, we chose an overall input rotation of 9°, which was reported as an average rotation of the meral-V (*Patek et al., 2007*) and we used minimum KT over this range for subsequent analyses. In sunfish, the overall rotation was set at 5°. We chose this angle because it included the majority of the four-bar rotation, including the point at which minimum KT was found in all but one species (*Micropterus coosae*). We found that patterns of mechanical sensitivity were not impacted by the use of dynamic or static measures of KT (*Supplementary file 16*).

## Mechanical sensitivity in four-bar linkage systems

To estimate mechanical sensitivity in each four-bar linkage system, we measured the relationship between KT and morphology (link length). To account for differences in scale, we log-transformed all traits prior to analyses (*Gingerich, 2009*; *O'Meara et al., 2006*; *Ackerly, 2009*; *Adams, 2013*). Relatedness is assumed to result in similar residuals from least-squares regressions, indicating non-independence of data points (*Felsenstein, 1985*). However, the extent to which phylogeny impacts the covariance structure of the residuals can vary substantially (*Revell, 2010*). To account for this, we employed a phylogenetic generalized least squares (PGLS) analysis, in which the maximum likelihood estimate of phylogenetic signal (λ) in the residual error is simultaneously estimated with the regression parameters. This method outperforms other approaches (including non-phylogenetic approaches) under a wide range of conditions (*Revell, 2010*). Regressions were performed using the *pgls* function in the R package caper with KT as the response variable and size-corrected linkages as the predictor variables (*Orme et al., 2012*; *R Core Development Team, 2014*). Following established methods in fish (*Westneat, 1990*; *Westneat, 1994*) and mantis shrimp (*Anderson et al., 2014*; *Anderson and Patek, 2015*; *Muñoz et al., 2017*), size-independent linkage measurements were calculated by dividing the output, input, and coupler links by the length of the fixed link. Nonetheless, estimates of mechanical sensitivity were robust to alternative size corrections (*Supplementary file 17*).

To visualize variation in mechanical sensitivity among links, we created rotatable 3D phylomorphospace plots for each system (*phylomorphospace3d* function, phytools package, *Revell, 2010*; interactive HTML plots: *rglwidget* function in the rgl package (https://r-forge.r-project.org/projects/rgl/).

## Testing for differences in evolutionary rates (tempo) and trait shifts (mode)

We estimated and compared the Brownian Motion (BM) rate parameter ($\sigma^2$) for the output, input, and coupler links using a likelihood ratio test. Specifically, we compared the likelihood of a model in which $\sigma^2$ varied among traits to one in which the rates were constrained to be equal (*Adams, 2013*). Because all traits are linear and were log-transformed, differences in evolutionary rates represent the amount of relative change in proportion to the mean and can be statistically compared (*O'Meara et al., 2006*; *Ackerly, 2009*; *Adams, 2013*). We bounded our estimates of $\sigma^2$ using a 95% confidence interval, which we derived from the standard errors of evolutionary rate. We obtained standard errors from the square root diagonals of the inverse Hessian matrix, using code provided by D. Adams. We then fitted a model in which rates of link evolution were constrained to be equal

(null hypothesis: $\sigma^2_{input} = \sigma^2_{output} = \sigma^2_{coupler}$) and a model in which evolutionary rates were free to vary among links ($\sigma^2_{input} \neq \sigma^2_{output} \neq \sigma^2_{coupler}$), and compared the models using likelihood ratio tests (*Adams, 2013*). We performed the same rate comparisons on every pairwise combination of mobile links. To connect evolutionary rate differences to mechanical sensitivity, we calculated the correlation between KT and link length for each system using phylogenetic generalized least squares regression (PGLS).

Higher variance in a trait can artificially inflate its estimated evolutionary rate (*Ives et al., 2007*; *Adams, 2013*). In both mantis shrimp and sunfish, the output link had more variance than the input and coupler links, which could have impacted its accelerated evolutionary rate relative to the other traits. Thus, we repeated the evolutionary rate comparisons while explicitly incorporating within-species measurement error using the ms.err option with the *compare.rates* function (*Adams, 2013*) (*Supplementary file 2*).

To test whether phylogenetic transitions in mechanical output are associated with the mechanical sensitivity of the system, we statistically assessed the number and location of strongly supported (i.e. high posterior probability) evolutionary shifts in all traits. We employed a reversible-jump Bayesian approach to test the fit of an Ornstein-Uhlenbeck (OU) model with one or more shifts in the trait optimum parameter ($\theta$) using the *bayou.mcmc* function in the R package bayou v.1.0.1 (*Hansen, 1997*; *Butler and King, 2004*; *Uyeda and Harmon, 2014*). The inference of evolutionary trait shifts is problematic when fewer than 50 species are included (*Uyeda and Harmon, 2014*); hence we only performed this analysis on wrasses (101 species). The method employs Markov chain Monte Carlo (MCMC) to sample the number, magnitudes, and locations of evolutionary shifts in the trait optimum on the time-calibrated phylogeny. Priors were defined such that they allowed a maximum of one shift per branch and equal probability among branches. For each trait, we performed two replicate analyses of two million MCMC generations each; for every analysis, we discarded the first 30% as burn-in. We assessed run convergence using Gelman's *R*-statistic (*Gelman and Rubin, 1992*) and visual comparison of likelihood traces. We estimated the location (branch number and position along branch) and the posterior probabilities of shifts for each trait.

## Acknowledgments

We thank Patek Lab members, D Adams, L Harmon, L Revell, J McGlothlin, J Uyeda, D McShea and two reviewers for feedback. Funding was provided by a National Science Foundation grant to SNP (IOS-1439850).

## Additional information

### Funding

| Funder | Grant reference number | Author |
|---|---|---|
| National Science Foundation | 1439850 | S N Patek |

The funders had no role in study design, data collection and interpretation, or the decision to submit the work for publication.

### Author contributions

Martha M Muñoz, Conceptualization, Data curation, Formal analysis, Supervision, Investigation, Visualization, Methodology, Writing—original draft, Writing—review and editing; Y Hu, Philip S L Anderson, Data curation, Writing—review and editing; SN Patek, Conceptualization, Resources, Supervision, Funding acquisition, Investigation, Methodology, Writing—original draft, Project administration, Writing—review and editing

### Author ORCIDs

SN Patek (ID) http://orcid.org/0000-0001-9738-882X

### Decision letter and Author response

Decision letter https://doi.org/10.7554/eLife.37621.028

Author response https://doi.org/10.7554/eLife.37621.029

## Additional files

### Supplementary files

• Supplementary file 1. Pair-wise comparisons of evolutionary rate. For each trait (KT, input link, output link, and coupler link), the Brownian Motion evolutionary rate parameter ($\sigma^2$) is given. For each comparison, the $AIC_C$ score for a model in which rates are allowed to vary (obs.) and constrained to be equal (const.) are given, as are the Likelihood Ratio Test (LRT) score and corresponding *p* value for df=1.

DOI: https://doi.org/10.7554/eLife.37621.009

• Supplementary file 2. Variance was substantially higher for the output link in mantis shrimp and sunfish. Therefore, for these systems, we repeated pair-wise comparisons of evolutionary rate while incorporating intraspecific measurement error (With Error). We also present results for rate comparisons without measurement error incorporated (No Error). For each comparison, the AIC score for a model in which rates are allowed to vary ($AIC_{observed}$) and constrained to be equal ($AIC_{constrained}$) are given, as are the Likelihood Ratio Test (LRT) score and corresponding *p* value for df=1. Evolutionary rate differences between linkages were robust to measurement error.

DOI: https://doi.org/10.7554/eLife.37621.010

• Supplementary file 3. A rotatable 3D phylomorphospace plot reveals variation in mechanical output (KT, denoted with color) with respect to oral four-bar linkage morphology (*x*, *y*, and *z* axes) across 30 species of cichlids (Family: Cichlidae). KT is inversely correlated with the input link and, to a lesser extent, positively correlated with the coupler link. To view these relationships, scroll the plot to place the input and coupler links as *x* and *y*-axes, and note the linear correlation between the two variables, coupled with a distinct color transition in the KT color map overlaid on the data. To zoom in and out of the plot scroll upwards and downwards, respectively.

DOI: https://doi.org/10.7554/eLife.37621.011

• Supplementary file 4. Mechanical output (KT, denoted with color) varies with respect to oral four-bar linkage morphology (*x*, *y*, and *z* axes) across 101 species of wrasses (Family: Labridae). KT is strongly correlated with the input and output links, and these relationships can be viewed dynamically in 3D phylomorphospace by scrolling the plot to place the input and output links as *x* and *y* axes, and noting the linear correlation between the two variables, coupled with a distinct color transition in the KT colormap overlaid on the data. Patterns of mechanical sensitivity in the wrasse oral four-bar were similar in the opercular four-bar of sunfish, but quite different from the oral four-bar of cichlids, despite a common evolutionary origin. To zoom in and out of the plot scroll upwards and downwards, respectively.

DOI: https://doi.org/10.7554/eLife.37621.012

• Supplementary file 5. A rotatable 3D phylomorphospace plot reveals differential mechanical sensitivity between KT (denoted with color map) and linkage morphology (*x*, *y*, and *z* axes of plot) across 19 species of sunfish (Family: Centrarchidae). KT is inversely correlated with the output link and positively correlated with the input link. To view this relationship, place the input and output links as *x* and *y*-axes, and note the linear correlation between the two variables, coupled with a distinct color transition in the KT color map overlaid on the data. The weak relationship association between the coupler link and KT can be viewed by placing the coupler link on the *x*-axis, and noting the lack of correlation with other links as well as no consistent association with the KT color map. This pattern of sensitivity is similar to the oral four-bar of wrasses.

DOI: https://doi.org/10.7554/eLife.37621.013

• Supplementary file 6. A rotatable 3D phylomorphospace illustrates variation in mechanical output (KT, denoted with color map) with respect to linkage morphology (*x*, *y*, and *z* axes of plot) across 36 species of mantis shrimp (Order: Stomatopoda). KT is inversely correlated with the output link, and is uncorrelated with the input and coupler links. These patterns can be visualized by rotating the plot to place the output link on the *x*-axis, and noting the strong transition from high values of KT (red) when the output link is small to low values of KT (blue) when the output link is large. Plotting

the input and coupler links on the *x* and *y* axes shows no relationship (color is dispersed across morphospace). Scrolling up and down zooms in and out of the plot, respectively.
DOI: https://doi.org/10.7554/eLife.37621.014

• Supplementary file 7. Analysis of evolutionary shifts for one mechanical trait (KT) and three morphological traits (input link, output link, and coupler link) in wrasses. For each shift, the branch and the shift's posterior probability (pp) are given. The branches that were strongly supported (pp > 0.5) are denoted in bold, and depicted in *Figure 3*.
DOI: https://doi.org/10.7554/eLife.37621.015

• Supplementary file 8. Time-calibrated phylogeny of cichlids (Family: Cichlidae) used in this study. This phylogeny was constructed using the previously-published phylogeny by *Hulsey et al., 2010*.
DOI: https://doi.org/10.7554/eLife.37621.016

• Supplementary file 9. Time-calibrated phylogeny of mantis shrimp (Stomatopoda) used in this study was based on the previously-published phylogeny by *Porter et al. (2010)*.
DOI: https://doi.org/10.7554/eLife.37621.017

• Supplementary file 10. Time-calibrated phylogeny of sunfish (Family: Centrarchidae) used in this study. We constructed this phylogeny using the previously-published phylogeny by *Near et al., 2005*.
DOI: https://doi.org/10.7554/eLife.37621.018

• Supplementary file 11. Time-calibrated phylogeny of wrasses (Family: Labridae) used in this study. This phylogeny is based on the previously-published wrasse phylogeny by *Baliga and Law, 2016*.
DOI: https://doi.org/10.7554/eLife.37621.019

• Supplementary file 12. The mechanical and morphological data for the cichlid (Family: Cichlidae) species used in this study. Data were gathered from *Hulsey and Garcia De Leon, 2005*.
DOI: https://doi.org/10.7554/eLife.37621.020

• Supplementary file 13. The mechanical and morphological data for the stomatopod species used in this study. Data were gathered from *Anderson and Patek, 2015*.
DOI: https://doi.org/10.7554/eLife.37621.021

• Supplementary file 14. The mechanical and morphological data for the sunfish (Family: Centrarchidae) species used in this study. Data were gathered from *Hu et al., 2017*.
DOI: https://doi.org/10.7554/eLife.37621.022

• Supplementary file 15. The mechanical and morphological data for the wrasses (Family: Labridae) used in this study. Data were gathered from *Alfaro et al., 2005*.
DOI: https://doi.org/10.7554/eLife.37621.023

• Supplementary file 16. Analyses of mechanical sensitivity in mantis shrimp and sunfish are robust to static measures of KT (see Materials and methods). Table shows PGLS regressions examining the relationship between mobile links (predictor variable) and kinematic transmission (response variable) using static (rather than dynamic) measures of KT in mantis shrimp and sunfish.
DOI: https://doi.org/10.7554/eLife.37621.024

• Supplementary file 17. Using residuals of trait values produces similar mechanical sensitivity results to the commonly utilized size corrections (mobile lever/fixed lever). Here, the predictor variables were the residuals of mobile link length regressed against fixed link length. Raw data for the wrasses were not available.
DOI: https://doi.org/10.7554/eLife.37621.025

• Transparent reporting form
DOI: https://doi.org/10.7554/eLife.37621.026

## Data availability
All datasets and phylogenies are included in full in the supplementary materials. Citations to the original papers containing these datasets and phylogenies are included with the supplementary files.

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
