## [Decision Letter]

Thank you for submitting your article "Strong mechanical relationships bias the tempo and mode of morphological evolution" for consideration by *eLife*. Your article has been reviewed by three peer reviewers, one of whom is a member of our Board of Reviewing Editors, and the evaluation has been overseen by Diethard Tautz as the Senior Editor. The following individual involved in review of your submission has agreed to reveal his identity: Peter Wainwright (Reviewer #2).

The reviewers have discussed the reviews with one another and the Reviewing Editor has drafted this decision to help you prepare a revised submission. Please respond in a point-by-point fashion and mark changes in the manuscript using a blue font. This will enable us to quickly decide if the authors complied and/or provided sufficient well-reasoned replies for publication, or that the discussion requires further reviewer consultation before a final decision can be made.

Summary:

The well-written manuscript of Muñoz et al., uses a key kinematic amplification mechanism in biomechanics, the enigmatic four-bar mechanism, to probe the tempo and mode of morphological evolution in four aquatic families (three fish and one mantis shrimp family). The compelling phylogenetic analysis shows how the mechanical sensitivity of the four-bar mechanism is correlated with rapid evolution. This key insight not only advances our understanding of how four-bar mechanisms evolved in these four families – the insightful outcome of this research is poised to encourage other comparative biomechanists to probe the mode and tempo of the evolution of other distinct biomechanical traits in organisms more generally. This research will not only interest comparative biomechanists, it may also engage workers in evolutionary biology, feeding ecology, and engineers working on bio-inspired mechanisms and robots.

Essential revisions:

1) Please clarify (A) in what way the analysis of evolutionary mode addresses issues not covered by the analysis of rate, so the reader understands the points you are trying to make. E.g. in both cases changes in KT seem to be mostly the result of changes in the more sensitive links in the four-bar. (B) How are these changes somehow qualitatively different? What the bayou analysis shows is that particularly big jumps in KT value are also mainly due to changes in the sensitive links, just as smaller changes in KT are mostly the result of changes in sensitive links. Hopefully this clarifies that it is difficult to understand the distinction between what the authors call 'rate' and 'mode' of evolution that is drawn in the Title, Abstract and Introduction. Please either adjust the text to clarify this and/or revise the title to draw attention to the main point the reader should focus on.

2) Another concern involves the analysis of mode, because of the apparent lack of independence of KT and the link lengths. KT is calculated directly from the link lengths, so how could a jump in KT (peak shift) not be associated with a jump in the traits that most strongly affect KT in the four-bar calculations? Please further consider the viability of the null hypothesis in your reply in light of how changes are connected via the relations between the links.

3) Considering the fixed link is an important component to the definition of the four-bar, and considering it is fixed, one may hypothesize it has the slowest rate of evolution. If so, it could potentially act as a control. Could the authors show that the fixed link covaries little with the rest of the four-bar system? Or better discuss why not so it is easier to understand the evolution of each link.

4) Please better explain the role of the four-bar mechanism in the ecomorphology for each family in the manuscript.

5) Introduction: Please clarify the 'mechanical output' that is specifically analyzed here; e.g. stroke/displacement amplification, force amplification, power amplification etc. The explanation in the methods section is helpful, the specific mechanical parameter analyzed should be clear in each section as well as the functional differences between the families. It would be great to highlight in the introduction if a family would 'care more' about displacement, velocity, force, power etc. Generalizations can follow in the discussion / conclusion section. A short and clear introduction / explanation of how one can be optimized versus the other will benefit general readers not familiar with biomechanics (you can then reference to the methods for details that the specialist reader would care about).

6) Introduction: 'transmit motion', this is incomplete, because also force or power can be transmitted / amplified.

7) Results section: 'smallest links', as you discuss later in the paragraph these require the smallest changes for a significant effect. In the Materials and methods section, it becomes clear you also looked carefully at the tempo of link length change and compared it to body size as well, in addition to kinematic transmission (KT). Overall it is not as clear in the main manuscript that you carefully looked at these aspects, please introduce them and clarify how the outcomes drive your conclusions. This will make it easier to follow the key findings and comprehend the extensive testing that support your thesis. (The same point holds for Discussion section; how about the mechanical sensitivity requiring minimal relative length changes, hence giving a high tempo? Please rewrite so the reader can easily grasp your point.)

8) Results section: Why was this not tested for the smaller datasets too, maybe just clarify that here more specifically?

9) Figure 2: The vertical axes are inconsistent and don't allow the reader to quickly grasp key differences in evolutionary rates. Please adjust for clarity along the following lines: Start all axis at 0. If you would let the axis end at 6 for the Wrasses and Sunfish (with a tick mark spacing of 1) the graph would be very intuitive and clear. If you want to zoom in a bit more on the Mantis, Shrimps, and Cichlids we can recommend making both axis start again at 0 and end at 3 instead, that way all results can still be compared easily (while zooming a bit in on the families with lower rates). Axis have to start at zero to ensure we are not zooming in on the variation/noise to see differences that are statistically significant but not large enough to matter strongly biologically / mechanically. By including zero and making axes ranges identical or a simple multiple (without many differences), the reader can estimate the% differences in rate within one glance instead of having to rescale each and every graph in his or her mind.

---

## [Author Response]

Essential revisions:1) Please clarify (A) in what way the analysis of evolutionary mode addresses issues not covered by the analysis of rate so the reader understands the points you are trying to make. E.g. in both cases changes in KT seem to be mostly the result of changes in the more sensitive links in the four-bar. (B) How are these changes somehow qualitatively different? What the bayou analysis shows is that particularly big jumps in KT value are also mainly due to changes in the sensitive links, just as smaller changes in KT are mostly the result of changes in sensitive links. Hopefully this clarifies that it is difficult to understand the distinction between what the authors call 'rate' and 'mode' of evolution that is drawn in the Title, Abstract and Introduction. Please either adjust the text to clarify this and/or revise the title to draw attention to the main point the reader should focus on.

We have added a figure (Figure 4) to illustrate the distinction, in addition to extensive text clarifications throughout the manuscript.

A) Evolutionary mode: Evolutionary mode examines the phylogenetic pattern of trait shifts. The bayou analysis offers a quantitative/statistical analysis of concordant patterns of evolutionary change (mode) in link length and KT, particularly in light of the mechanical sensitivity of the links. This is different than the analysis of evolutionary rates that assesses the rates of change of KT and link lengths, but “black-boxes” whether those changes happened at the same points on the phylogeny, and whether or not they occurred in the links to which the system is most mechanically sensitive.

B) Specifically, in the bayou analysis, we asked whether the links to which the system is most mechanically sensitive shifted concordantly with KT. These shifts could have happened at different times or places on the phylogeny, or via multiple changes of different link identities. By statistically identifying the significant shifts in KT and link length across the tree topology, we discovered that shifts in KT occurred at the same places on the tree as the links to which the system is most mechanically sensitive links. Moreover, we found that there were multiple morphological pathways (exclusively in these links) accompanying transitions in KT. This finding addresses the mode (pattern) of evolution rather than tempo (rate). Our results illustrate how mechanical sensitivity and many-to-one mapping are connected in the evolution of mechanical structures.

2) Another concern involves the analysis of mode, because of the apparent lack of independence of KT and the link lengths. KT is calculated directly from the link lengths, so how could a jump in KT (peak shift) not be associated with a jump in the traits that most strongly affect KT in the four-bar calculations? Please further consider the viability of the null hypothesis in your reply in light of how changes are connected via the relations between the links.

At the level of geometric calculations, KT should change with the components to which it is most sensitive. At the level of evolutionary rates analysis (tempo), this also occurs in the form of shared acceleration of rates. We can examine this question at a third level – the pattern of trait changes across the topology of the tree (mode) – which offers another lens on the arena of mechanical sensitivity in that the traits are changing concordantly (i.e., together, in the same place on the tree). Each of these levels of analysis is necessary to fully integrate and explain the biomechanics-evolution interface. Simply because two traits change together due to physics, need not be manifested as shared accelerated rate or even shared evolutionary dynamics across a phylogeny. As described in the response above, the analysis of mode provides a new perspective on the connection between mechanical sensitivity and many-to-one mapping that the rate analysis cannot address. We address these issues throughout the revised manuscript.

3) Considering the fixed link is an important component to the definition of the four-bar, and considering it is fixed, one may hypothesize it has the slowest rate of evolution. If so, it could potentially act as a control. Could the authors show that the fixed link covaries little with the rest of the four-bar system? Or better discuss why not so it is easier to understand the evolution of each link.

The fixed link is used in all four-bar linkage analyses to size-standardize the mobile links, a step that is necessary for performing any evolutionary analysis. Using a different metric for size standardization is not possible given that the previously published data do not include other measurements for size standardization, making the fixed link the only choice and the choice most consistent with previous research. Therefore, even though this is an important consideration, it is not presently possible to implement it in four-bar linkage systems – although it could be addressed in other mechanical systems in future studies.

4) Please better explain the role of the four-bar mechanism in the ecomorphology for each family in the manuscript.

We have edited the manuscript to address this comment, particularly in the Introduction and Discussion section.

5) Introduction: Please clarify the 'mechanical output' that is specifically analyzed here; e.g. stroke/displacement amplification, force amplification, power amplification etc. The explanation in the methods section is helpful, the specific mechanical parameter analyzed should be clear in each section as well as the functional differences between the families. It would be great to highlight in the introduction if a family would 'care more' about displacement, velocity, force, power etc. Generalizations can follow in the discussion / conclusion section. A short and clear introduction / explanation of how one can be optimized versus the other will benefit general readers not familiar with biomechanics (you can then reference to the methods for details that the specialist reader would care about).

In our revision we clarify the mechanical importance of KT, how it reflects a tradeoff between displacement and force, and how those tradeoffs, in a broad sense, align with trophic ecology. We chose not to discuss in detail, however, optimization of KT in a biomechanical sense because this a question that is distinct from the questions we address in this study. It can also have the unfortunate consequence of causing confusion with respect to our evolutionary analysis of the parameter called the evolutionary trait optimum (theta) (which is not a measure of biomechanical optimization – an unfortunate terminological confusion that regularly arises when the field of phylogenetic analysis integrates with biomechanics). We have noted the distinction between biomechanical optimization and the evolutionary trait parameter theta in the Materials and methods section.

6) Introduction: 'transmit motion', this is incomplete, because also force or power can be transmitted / amplified.

We have changed the text to “transmit motion and force”.

7) Results section: 'smallest links', as you discuss later in the paragraph these require the smallest changes for a significant effect. In the Materials and methods section, it becomes clear you also looked carefully at the tempo of link length change and compared it to body size as well, in addition to kinematic transmission (KT). Overall it is not as clear in the main manuscript that you carefully looked at these aspects, please introduce them and clarify how the outcomes drive your conclusions. This will make it easier to follow the key findings and comprehend the extensive testing that support your thesis. (The same point holds for Discussion section; how about the mechanical sensitivity requiring minimal relative length changes, hence giving a high tempo? Please rewrite so the reader can easily grasp your point.)

We have now addressed the issue of link size throughout the manuscript, including a new paragraph in the Discussion section.

8) Results section: Why was this not tested for the smaller datasets too, maybe just clarify that here more specifically?

We did not test this in the other clades, because the analysis is unstable when there are fewer than 50 taxa (Uyeda and Harmon, 2014), which we confirmed when tried to pilot the analyses in our smaller datasets. We mention this in the revised manuscript.

9) Figure 2: The vertical axes are inconsistent and don't allow the reader to quickly grasp key differences in evolutionary rates. Please adjust for clarity along the following lines: Start all axis at 0. If you would let the axis end at 6 for the Wrasses and Sunfish (with a tick mark spacing of 1) the graph would be very intuitive and clear. If you want to zoom in a bit more on the Mantis, Shrimps, and Cichlids we can recommend making both axis start again at 0 and end at 3 instead, that way all results can still be compared easily (while zooming a bit in on the families with lower rates). Axis have to start at zero to ensure we are not zooming in on the variation/noise to see differences that are statistically significant but not large enough to matter strongly biologically / mechanically. By including zero and making axes ranges identical or a simple multiple (without many differences), the reader can estimate the% differences in rate within one glance instead of having to rescale each and every graph in his or her mind.

We edited the figure per the suggestions above, and changed the y-axis go to nine in cichlids so that all traits could fit in the figure.